# α-Melanocyte-Stimulating Hormone Attenuates Neovascularization by Inducing Nitric Oxide Deficiency via MC-Rs/PKA/NF-κB Signaling

**DOI:** 10.3390/ijms19123823

**Published:** 2018-11-30

**Authors:** Wen-Tsan Weng, Chieh-Shan Wu, Feng-Sheng Wang, Chang-Yi Wu, Yi-Ling Ma, Hoi-Hung Chan, Den-Chiung Wu, Jian-Ching Wu, Tian-Huei Chu, Shih-Chung Huang, Ming-Hong Tai

**Affiliations:** 1Department of Medical Research, Kaohsiung Chang Gung Memorial Hospital, Kaohsiung 83301, Taiwan; bpvincent@gmail.com (W.-T.W.); wangfs@ms33.hinet.net (F.-S.W.); 2Core Facility for Phenomics & Diagnostics, Kaohsiung Chang Gung Memorial Hospital, Kaohsiung 83301, Taiwan; 3Department of Dermatology, Kaohsiung Veterans General Hospital, Kaohsiung 81362, Taiwan; dermawu@vghks.gov.tw; 4Graduate Institute of Clinical Medical Sciences, Chang Gung University College of Medicine, Kaohsiung 83301, Taiwan; 5Department of Biological Sciences, National Sun Yat-sen University, Kaohsiung 80424, Taiwan; cywu@mail.nsysu.edu.tw; 6Division of Nephrology, Kaohsiung Veterans General Hospital, Kaohsiung 81362, Taiwan; ylma@vghks.gov.tw; 7Division of Gastroenterology, Department of Internal Medicine, Kaohsiung Veterans General Hospital, Kaohsiung 81362, Taiwan; hoihungchan@gmail.com; 8Center for Stem Cell Research, Kaohsiung Medical University Hospital, Kaohsiung 80708, Taiwan; wejewu@cc.kmu.edu.tw; 9Division of Gastroenterology, Department of Internal Medicine, Kaohsiung Medical University Hospital, Kaohsiung 80708, Taiwan; 10Doctoral Degree Program in Marine Biotechnology, National Sun Yat-sen University and Academia Sinica, Kaohsiung 804, Taiwan; djbluestyle338@hotmail.com; 11Center for Neuroscience, National Sun Yat-sen University, Kaohsiung 80424, Taiwan; skbboyz0817@gmail.com; 12Institute of Biomedical Sciences, National Sun Yat-sen University, Kaohsiung 80424, Taiwan; 13Department of Internal Medicine, Kaohsiung Armed Forces General Hospital, Kaohsiung 80284, Taiwan; 14Graduate Institute of Medicine, Kaohsiung Medical University, Kaohsiung 80708, Taiwan

**Keywords:** α-melanocyte-stimulating hormone (α-MSH), melanocortin receptors (MC-Rs), HUVECs, nitric oxide (NO)

## Abstract

α-melanocyte-stimulating hormone (α-MSH) has been characterized as a novel angiogenesis inhibitor. The homeostasis of nitric oxide (NO) plays an important role in neovascularization. However, it remains unclear whether α-MSH mitigates angiogenesis through modulation of NO and its signaling pathway. The present study elucidated the function and mechanism of NO signaling in α-MSH-induced angiogenesis inhibition using cultured human umbilical vein endothelial cells (HUVECs), rat aorta rings, and transgenic zebrafish. By Griess reagent assay, it was found α-MSH dose-dependently reduced the NO release in HUVECs. Immunoblotting and immunofluorescence analysis revealed α-MSH potently suppressed endothelial and inducible nitric oxide synthase (eNOS/iNOS) expression, which was accompanied with inhibition of nuclear factor kappa B (NF-κB) activities. Excessive supply of NO donor l-arginine reversed the α-MSH-induced angiogenesis inhibition in vitro and in vivo. By using antibody neutralization and RNA interference, it was delineated that melanocortin-1 receptor (MC1-R) and melanocortin-2 receptor (MC2-R) participated in α-MSH-induced inhibition of NO production and NF-κB/eNOS/iNOS signaling. This was supported by pharmaceutical inhibition of protein kinase A (PKA), the downstream effector of MC-Rs signaling, using H89 abolished the α-MSH-mediated suppression of NO release and eNOS/iNOS protein level. Therefore, α-MSH exerts anti-angiogenic function by perturbing NO bioavailability and eNOS/iNOS expression in endothelial cells.

## 1. Introduction

α-Melanocyte-stimulating hormone (α-MSH) is a 13-amino acid neuropeptide derived from the precursor proopiomelanocortin (POMC), a pivotal stress hormone of hypothalamus/pituitary/adrenal gland (HPA) axis [1]. α-MSH is widely expressed in various tissues including hypothalamus, skin, and immune cells [2,3]. Cumulative evidence reveals that pharmacological α-MSH administration alleviates inflammatory reaction of acute respiratory distress syndrome, sepsis, rheumatoid arthritis, and inflammatory bowel disease [4,5,6]. In rodent models, α-MSH attenuates the autoimmune uveitis-mediated inflammation, shields photoreceptors from retinal dystrophy, and mitigates the diabetes-mediated retinal vessel damage [7,8]. Application of the synthetic α-MSH analog, Afamelanotide ([Nle4-D-Phe7]-α-MSH), alleviates dermatological disorders, including polymorphic light eruption, erythropoietic protoporphyria, and vitiligo [9,10]. Besides, α-MSH also confers protection against neurodegenerative diseases and cerebral ischemia [11,12,13,14]. Moreover, recent studies have demonstrated the anti-neoplastic potential of POMC, the precursor of α-MSH, in various types of cancer through neovascularization blockade [15,16,17]. Thus, the anti-angiogenic function of α-MSH is highly relevant to clinical science and could be applicable to diseases due to excessive neovascularization.

There are five G-protein-coupled melanocortin receptors (MC1-R to MC5-R) which are known to transmit the signaling of melanocortins including α-MSH [1]. For example, MC1-R mediates the effect of α-MSH on melanogenesis, thermoregulation, and pigmentation formation [18]. Activation of MC4-R and MC5-R contribute to the α-MSH modulation of food intake, energy expenditure, and fatty acid oxidation within skeletal muscle [19,20]. In addition to melanocytes and cells in the central nervous system, MC-Rs also modulate the cellular function and behaviors of various types of cells including inflammatory cells, keratinocytes, fibroblasts, adipocytes, and endothelial cells [21].

Angiogenesis plays a critical role in tissue development, remodeling, and tumorigenesis in physiological and pathological conditions such as psoriasis and cancer [22,23]. The endogenous angiogenesis switch is delicately regulated by the reciprocal interaction between endogenous angiogenic factors and inhibitors. Our previous study unveiled the anti-angiogenic function of α-MSH in vitro and in vivo [21]. Besides, vascular endothelial growth factor (VEGF) depletion and blockage of VEGF receptor-2 (VEGFR2)/Akt signaling was involved in α-MSH-mediated angiogenesis inhibition. However, treatment with excess VEGF partially alleviated the α-MSH-induced neovascularization blockade [21], implying that α-MSH may regulate the bioavailability of additional angiogenic factor(s). Among the endogenous pro-angiogenic factors, nitric oxide (NO) synthesized by nitric oxide synthases (NOSs) is an important regulator of angiogenesis in either physiological or pathological contexts [24]. NO stimulates the proliferation and migration of endothelial cells through activation of cGMP and mitogen-activated kinase (MAPK) signaling [25]. Besides, NO also participates in the VEGF-mediated enhancement of endothelial sprouting through PI3K-Akt pathways [25]. Overabundant NO is observed to trigger the vascularization in inflammatory tissues [15], suggesting the indispensability of NO in initiating angiogenesis. It was hypothesized that NO may be involved in the α-MSH mitigation of vessel morphogenesis. Thus, this study aims to elucidate the role and mechanism of NO/NOSs signaling in α-MSH-induced angiogenesis inhibition.

## 2. Results

### 2.1. Nitric Oxide (NO) Deficiency Participated in α-Melanocyte-Stimulating Hormone (α-MSH)-Induced Angiogenesis Inhibition

By using the Griess reagent to determine the nitrite level as an index of NO production [26], we evaluated the influence of α-MSH on NO production and found that α-MSH within physiological concentrations (0.01–10 nM) dose-dependently reduced the NO production in Human umbilical vein endothelial cells (HUVECs) (Figure 1A). We then investigated whether supply of endogenous NO donor l-arginine (l-Arg) affected the anti-angiogenic function of α-MSH. It was revealed that l-Arg supplementation significantly ameliorated the α-MSH-mediated suppression of migration (Figure 1B) and tube formation in HUVECs (Figure 1C). Furthermore, l-Arg application significantly restored microvessel sprouting in α-MSH-treated aortic explants (Figure 1D). These results indicated that defective NO homeostasis contributed to α-MSH-induced angiogenesis inhibition in vitro and ex vivo.

### 2.2. l-Arginine (l-Arg) Supply Mitigated the α-MSH-Induced Neovascularization Blockade in Zebrafish

To verify the role of NO deficiency in α-MSH-mediated angiogenesis suppression in vivo, we employed the double transgenic *Tg*(*kdrl:mCherry^ci5^; fli1a:negfp^y7^*) zebrafish with mCherry-tagged intersegmental vessels (ISV) in red and enhanced green fluorescent protein (EGFP)-expression in nucleus of endothelial cells [27]. After treatment with α-MSH with or without l-Arg for 24 h (at 30 hpf), all zebrafish embryos remained viable and displayed normal morphologies (Figure 2A). By quantitative analysis, it was found that α-MSH treatment potently perturbed the vessel development in zebrafish, which exhibited IVS hypotrophy (13.3% completion in α-MSH group versus 98.3% completion in PBS control; Figure 2B) with fewer endothelial cells per vessel (1.9 ± 0.8 cells in α-MSH group versus 3.9 ± 0.7 cells per ISV in PBS control; Figure 2C). Despite the lack of effect on vessel development, l-Arg supplementation significantly rescued the α-MSH-induced ISV defects (91.2% completion; Figure 2B) and restored endothelial recruitment in ISV (3.5 ± 0.7 endothelial cells per ISV; Figure 2C). Therefore, l-Arg supply rescued the α-MSH-impeded vessels development in vivo.

### 2.3. NO/Soluble Guanylyl Cyclase (sGC) Signaling Modulates α-MSH-Induced Angiogenesis Inhibition

Since soluble guanylyl cyclase (sGC) is the downstream effector of NO signaling [28], we utilized the synthetic NO donors including nitroglycerin (NTG) and sodium nitroprusside (SNP) [29,30] to modulate sGC activities then monitored their influences on α-MSH-induced angiogenesis inhibition. It was shown that treatment with either NTG or SNP significantly attenuated the inhibition of α-MSH on tube formation in HUVECs (Figure 3A). Moreover, in transgenic zebrafish, SNP supply significantly alleviated the α-MSH-attenuated ISV development and enhanced the recruitment of endothelial cells in ISV (Figure 3B). Quantification analysis revealed that SNP supply enhanced the ISVs completion (98.7% compared with 13.0% in α-MSH group) and the number of migrated endothelial cells per ISVs (3.9 ± 0.7 cells compared with 1.9 ± 0.8 cells in α-MSH group). Together, these results suggested that NO/sGC signaling indeed modulated the α-MSH-induced neovascularization blockade.

### 2.4. α-MSH Reduced Endothelial Endothelial and Inducible Nitric Oxide Synthase (eNOS and iNOS) Expression in a Dose-Dependent Manner

Because endothelial and inducible nitric oxide synthase (eNOS and iNOS) contribute to NO metabolism in endothelial cells [25], we examined the effect of α-MSH on eNOS phosphorylation and eNOS/iNOS expression in HUVECs. First, we observed that α-MSH significantly attenuated the eNOS (Figure 4A) and iNOS (Figure 4B) mRNA expression in HUVECs. By immunoblotting analyses, it was also found a 24-h α-MSH treatment elicited a dose-dependent reduction of eNOS phosphorylation and eNOS/iNOS protein levels in HUVECs (Figure 4C). Consistently, immunofluorescence analyses also revealed that α-MSH treatment depleted the expression of iNOS (Figure 4D), and eNOS (Figure 4E) in HUVECs. Time series analysis suggested that a 12-h treatment was required for α-MSH (at 10 nM) to achieve significant reduction of iNOS and eNOS expression in endothelial cells. Interestingly, α-MSH elicited a biphasic regulation on eNOS Ser1177 phosphorylation (increased in the initial 1 h then decreased after 8 h treatment) (Figure 4F). The above findings indicated that α-MSH decreased NO bioavailability in endothelial cells by diminishing eNOS and iNOS levels.

### 2.5. α-MSH Inhibited the Basal NF-κB Activities in Endothelial Cells by Elevating IκB and Reducing p65/p50/p105 Expression

Given that the NF-κB signaling pathway is involved in NOSs gene regulation [31], we examined whether α-MSH modulated the NOSs expression via regulating NF-κB signaling in endothelial cells. By using NF-κB-driven luciferase assay, it was observed that α-MSH treatment significantly attenuated the NF-κB promoter activities in either unstimulated or LPS-stimulated HUVECs by about 40% of control (Figure 5A). Immunoblot analysis further revealed that α-MSH potently and dose-dependently decreased the levels of NF-κB activation subunits, including p105, p65, and p50 (Figure 5B). On the contrary, α-MSH treatment increased the levels of NF-κB inhibitory subunit IκB while it reduced the levels of phosphorylated IκB (Figure 5C). Noteworthily, time series studies unveiled that α-MSH treatment rapidly elevated the IκB level in HUVECs (within 0.5 h) but required longer exposure (after 4-h) to achieve significant p65 downregulation (Figure 5D). These data indicated α-MSH induced inhibition of NF-κB signaling, thereby suppressing eNOS/iNOS expression in endothelial cells.

### 2.6. MC1-R and MC2-R Contributed to the α-MSH-Induced Reduction of eNOS and iNOS Signaling

There are four types of melanocortin receptors including MC1-R, MC2-R, MC4-R, and MC5-R in endothelial cells [21]. In our previous study, both MC-1R and MC2-R are required for α-MSH-mediated inhibition of VEGF/VEGFR2 signaling in endothelial cells [21]. By using an antibody neutralization strategy, it was shown that blocking either MC1-R or MC2-R restored the NO release in α-MSH-treated HUVECs (Figure 6A). By immunoblot analysis, it was found MC1-R neutralization abolished the α-MSH-induced eNOS/iNOS downregulation (Figure 6B). However, MC2-R neutralization reversed the α-MSH-mediated eNOS but not iNOS downregulation in HUVECs. Such notion was also supported by immunofluorescence analysis, in which application of MC1-R antibody prominently restored eNOS/iNOS protein levels, whereas MC2-R antibody rescued only eNOS expression in α-MSH-treated cells (Figure 6C,D). Hence, MC-1R and MC2-R participated in α-MSH-mediated regulation of eNOS/iNOS signaling in endothelial cells.

Since MC2-R is not a receptor for α-MSH, the role of MC2-R in mediating the function of α-MSH became paradoxical. Thus, RNA interference strategy was employed to validate the above finding. By immunoblot analysis, it was observed that prior transfection with MC1-R siRNA (siMC1-R) effectively alleviated the α-MSH-induced p65/iNOS/eNOS suppression, whereas MC2-R siRNA (siMC2-R) had no such effect (Appendix A). However, despite a lack of effect by itself, siMC2-R combination augmented the efficacy of siMC1-R in reverting the α-MSH-mediated p65/iNOS/eNOS inhibition in HUVECs (Appendix A). To further confirm such finding, we utilized NF-κB luciferase assay to evaluate the efficacy of different MC1-R/MC2-R-targeting approaches in alleviating the α-MSH-induced NF-κB suppression. It was shown that MC1-R intervention (by either antibody neutralization or RNA interference) was more potent than that of MC2-R in relieving the α-MSH-induced NF-κB inhibition (Appendix A). Nevertheless, simultaneous MC1-R and MC2-R blockage remained most effective in abolishing the α-MSH-induced NF-κB inhibition. Together, these results indicated the pivotal role of MC1-R and a relatively minor yet undeniable function of MC2-R in the mechanism underlying α-MSH-induced NF-κB inhibition in endothelial cells.

### 2.7. Pharmaceutical Protein Kinase A Inhibition by H89 Abolished the α-MSH-Induced NO Deficiency and Inhibition of NO/NOS Signaling

Since protein kinase A (PKA) is an important downstream effector of MC-Rs [32], we evaluated the contribution of PKA signaling in α-MSH-induced angiogenesis inhibition using tube formation assay. It was found that, similar to combined MC1-R and MC2-R antibody neutralization, PKA blockage by selective inhibitor H89 abolished the α-MSH-induced repression of tube formation in endothelial cells (Appendix A), implying that PKA plays a major role in α-MSH/MC1-R & MC2-R signaling. Consistently, H89 application effectively restored the NO production in α-MSH-treated HUVECs (Figure 7A). Besides, immunoblot analysis revealed that H89 reversed the α-MSH-induced inhibition of p65/iNOS/eNOS levels (Figure 7B). Finally, immunofluorescence analysis further confirmed the above findings in iNOS/NF-κB p65 (Figure 7C) and eNOS/phospho-Ser1177-eNOS immunoreactivities (Figure 7D). Together, PKA was the primary transducer of α-MSH/MC1-R/MC2-R signaling cascade in regulating NO homeostasis and NF-κB/NOSs activities.

## 3. Discussion

POMC is the precursor of melanocortins, including ACTH, α-MSH, β-MSH, and γ-MSH, which are well known for their multifaceted actions in control of steroidogenesis, skin pigmentation, energy homeostasis, and inflammation [1,33,34]. The inhibitory function of POMC on neovascularization was first discovered in cultured endothelial cells, in which POMC overexpression potently suppressed the angiogenic processes including migration and tube formation of endothelial cells [35]. In recent years, we have demonstrated the anti-angiogenic potential of POMC therapy in pre-clinical animal models of diseases due to excessive angiogenesis such as B16-F10 melanoma [15,16], Lewis lung carcinoma [16], and osteoarthritis [36]. Subsequently, α-MSH, one of the POMC-derived melanocortins, has been characterized to play a critical role in POMC-mediated neovascularization blockade through downregulation of VEGF/VEGFR2 signaling pathway [21]. However, restoring VEGF/VEGFR2 axis using excessive VEGF or VEGFR2 agonist failed to relieve the α-MSH-induced neovascularization blockade. Therefore, this study elucidated the involvement of NO/NOSs signaling in the anti-angiogenic mechanism of α-MSH. Indeed, the present study has unveiled that α-MSH at physiological concentrations (at as low as 0.01 nM) effectively attenuates the NO secretion of unstimulated endothelial cells. Moreover, exogenous supply of endogenous NO donor l-Arg or synthetic SNP potently rescued the α-MSH-induced neovascularization blockade in vitro and in vivo. Finally, the α-MSH-mediated NO deficiency was delineated through inhibition of NF-ΚB/eNOS/iNOS cascade in endothelial cells. Thus, NO/NOSs signaling is pivotal to the anti-angiogenic function of α-MSH.

NO regulates angiogenesis and vascular permeability in various tissues under physiological and pathological contexts [22]. The present study clearly demonstrates application of α-MSH suppresses NO signaling and activities in endothelial cells and developing vascular systems. However, the effects of α-MSH on NO homeostasis in other tissues remain controversial [37]. In spontaneously hypertensive rats, microinjection of α-MSH into the nucleus tractus solitarii in the brain stem promptly elicits a hypotensive effect through NO release by iNOS activation [37]. Likewise, in vessel relaxation assay, application of α-MSH at high dose for short time interval (1 μM for 1 h) increases the NO burst and eNOS phosphorylation in endothelial cells of aorta vessels [38]. Due to the limitation of NO detection assay used, the kinetics of NO level in α-MSH-treated endothelial cells could not be precisely determined. Nevertheless, the time series study revealed a biphasic effect of α-MSH on Ser1177 eNOS phosphorylation (Figure 4F), in which eNOS phosphorylation significantly elevated within 0.5–1 h after α-MSH treatment then declined to lower than basal level after 8 h. Interestingly, eNOS Ser1177 can be phosphorylated by PKA and its phosphorylation is correlated with eNOS activation [39]. Therefore, it seemed plausible that α-MSH may increase NO production via PKA-mediated eNOS phosphorylation in the early phase. However, longer α-MSH exposure (>8 h) led to NO deficiency by transcriptional repression of eNOS/iNOS level via suppression of NF-ΚB signaling. Future studies are warranted to explore the relationship between α-MSH and NO homeostasis in different cells types and tissues/organs.

By employing antibody neutralization, RNA interference, and luciferase assays, we have delineated the pivotal role of MC1-R and a relatively minor contribution of MC2-R, in the mechanism underlying α-MSH-induced NF-κB inhibition in endothelial cells. Furthermore, application of MC2-R antibody partially ameliorated the NO deficiency (Figure 6A), migration [21] and tube formation (Appendix A) in α-MSH-treated endothelial cells. The above findings support the essential role of MC2-R in mediating the anti-angiogenic function of α-MSH. However, such an important notion is paradoxical because MC2-R is known as an ACTH receptor that solely interacts with ACTH but no other melanocortins, including α-MSH [40]. Since MC1-R is capable of forming homodimers or heterodimers with other G-protein-coupled receptors during cellular activation [41], one likely explanation is that the presence of excessive α-MSH might stimulate the formation of MC1-R/MC1-R homodimers as well as MC1-R/MC2-R heterodimers, thereby maximally activating the downstream cAMP/PKA signaling in endothelial cells. Further studies are necessary to confirm the existence of MC1-R/MC2-R heterodimers in endothelial cells.

## 4. Materials and Methods

### 4.1. Peptides and Antibodies

α-MSH was purchased from Bachem Bioscience Inc. (King of Prussia, PA, USA). Calcein-AM was purchased from Molecular Probes (Thermo Fisher Scientific, Waltham, MA, USA), NOS substrate l-arginine (l-Arg), synthetic NO donors including sodium nitroprusside (SNP) and nitroglycerin (NTG), PKA inhibitor H89 were obtained from Sigma Chemical (St. Louis, MO, USA). α-MSH, l-Arg, SNP, NTG, and H89 were dissolved and diluted in sterile normal saline. Antibodies against eNOS, iNOS, and phosphorylated Ser1177 of eNOS (phosphor-Ser1177-eNOS) were obtained from BD Biosciences (San Diego, CA, USA). NF-κB p105, p50, p65, phosphorylated IκB (p-IκB), IκB, MC1-R, and MC2-R antibodies were obtained from Santa Cruz Biotechnology, Inc. (Santa Cruz, CA, USA); β-actin antibody was obtained from Sigma Chemical.

### 4.2. Endothelial Cells Cultures

Human umbilical vein endothelial cells (HUVECs) were obtained from Taiwan Medical Cell and Microbial Resource and incubated in M199 medium (Life Technologies, Gaithersburg, MD, USA), as previously described [42]. HUVECs within passages 2 to 5 were used for experiments.

### 4.3. Nitric Oxide Measurement

Nitric oxide release was quantified from the concentration of nitrite, a stable metabolite of NO, in culture media as previously described [42]. After treatment with α-MSH or a combination with 10 μM H89 in serum-free medium for 24 h, 50 μL of culture medium was mixed with 50 μL of Greiss reagent containing 0.1% naphthylethylenediamine dihydrochloride and 1% sulfanilamide in 5% phosphoric acid (*v*/*v*; 1:1). The absorbance of mixtures was spectrophotometrically measured at a wavelength of 540 nm (Novo Biolabs; Molecular Devices, San Jose, CA, USA). The concentrations of nitrite in culture media were calculated from a standard curve calibrated with various levels of sodium nitrite.

### 4.4. Boyden Chamber Assay

HUVECs were seeded in triplicate in the upper compartment of the Boyden chamber and supplemented with M199 media, as previously described [21]. The lower compartment of the chamber was filled with 30 µL of M199 media containing 10% FBS. Polycarbonate filters (8 µm pore size, Nucleopore; Costar, Cambridge, MA, USA) coated with 0.1% gelatin were layered between the compartments. Aliquots of 10 nM α-MSH and 5 mM l-arginine were added alone, or in combination, to the upper chamber. After incubation, the HUVECs migrated onto the lower side were fixed in absolute methanol and stained with 10% Giemsa solution (Merck, Germany). Number of cells of five different high-power fields were counted under a microscope and digital images system (Olympus; Tokyo, Japan) and expressed as mean ± SEM/field.

### 4.5. Tube Formation Assay

The tube formation of HUVECs was detected, as described previously [21]. Briefly, cell suspensions in M199 media containing 10% FBS were applied onto 96-well plates coated with Matrigel (10 mg/mL; BD Biosciences, San Diego, CA, USA). In some experiments, cell cultures were pretreated with 5 mM l-arginine, 10 μM NTG, 10 μM SNP, 10 μM H89, 10 μg/mL MC1-R antibody, and 10 μg/mL MC2-R antibody for 30 min or in combination with 10 nM α-MSH for 4 h at 37 °C in 5% CO_2_. Before observing the tube formation, calcein-AM (25 Μm) was applied into the medium and incubated for 30 min in the dark. The number of endothelial tubes in five different high-power fields was counted.

### 4.6. Aortic Ring Assay

The microvessel sprouting in rat aortic rings was performed as previously described [21]. Briefly, thoracic aorta was dissected from euthanatized 8-week-old Sprague–Dawley rats and followed by transverse section into the ring shape. Each aorta ring was embedded in the 1 mL mixtures of Matrigel and MCDB131 media (Life technologies Ltd., Paisley, Scotland, UK; 1:1, *v*/*v*). For assessment of vascular sprouting, each well was further added 1 mL MCDB131 medium containing 10 nM α-MSH or/combination with 5 mM l-arginine onto the Matrigel and incubated in a 37 °C, 5% CO_2_ chamber for 7 days. The length of vascular sprouts, the length from the aortic ring to the end of the greatest sprout was measured with the microscope and digital image system. Five fields in each aortic ring; 6 aortic rings in each group were randomly selected for quantification.

### 4.7. Angiogenesis in Zebrafish Model

Zebrafish were maintained at the 28.5 °C thermostatic aquaria which contained circulating, filtered, and aerated fresh water in a 14 h light/10 h dark cycle. Zebrafish embryos were nurtured in E3 embryo media at 28.5 °C, as previously described [27]. At 6 h post-fertilization (hpf), embryos were incubated in E3 media containing 20 mg/ml pronase (Sigma, St. Louis, MO, USA) to remove chorions; endogenous pigmentation was blocked by adding 0.003% *N*-phenylthiourea (Sigma, St. Louis, MO, USA). Embryos were incubated in 6-cm dishes containing 10 nM α-MSH or in combination with 5 mM l-arginine or 10 μM sodium nitroprusside. At 30 hpf, embryos were anesthetized, fixed, and embedded in 3% methylcellulose and subjected to microscopic observation using Zeiss Lumar V12 stereomicroscope and Zeiss Axiocam camera (Carl Zeiss, Oberkochen, Germany). Acquisition of confocal images was performed using Zeiss LSM510 or LSM700 microscope and Image J or ZEN 2012 software (Carl Zeiss, Oberkochen, Germany). The vascular development and the number of endothelial cells in intersegmental vessels (ISV) between the 5th and 15th ISVs from 3 embryos per group were counted. After completing experiments, the embryos were euthanatized with an overdose of 2-phenoylethanol.

### 4.8. Reverse Transcription-Quantitative Polymerase Chain Reaction (RT-qPCR)

HUVECs were lysed in a TRIzol reagent (TEL-TEST, Inc., Friendswoods, TX, USA) to collect total RNA. Five µg of the total RNA was used for the reverse transcription with Superscriptase III (Invitrogen, Carlsbad, CA, USA), oligo-dT, and random primers. The RT products were subjected to real-time PCR using SYBR Green PCR Master Mix and Lightcycler (Roche, Nederland, BV, USA), according to the protocols provided by the manufacturers. Sequences of primers for eNOS: forward 5′-TGAAGGCGACAATCCTGTATG-3′, reverse 5′-CTGCAAAGCTCTCTCCATTCT-3′; iNOS: forward 5′-GTGTTCCACCAGGAGATGTT-3′, reverse 5′-GTCTCAGTAGCAAAGAGGACTG-3′. Expression was normalized to β-actin: forward 5′-TCACCCACACTGTGCCCATCTACGA-3′, reverse 5′-CAGCGGAACCGCTC ATTGCCAATGG-3′.

### 4.9. Western Blot Analysis

Cell lysates were prepared using RIPA lysis buffer containing 0.25% sodium deoxycholate, 50 mM pH 7.4 Tris-HCl, 1% NP-40, 1 mM PMSF, 150 mM NaCl and protease inhibitors. Aliquot of lysates were subjected to electrophoresis and transferred onto the polyvinylidene difluoride membranes (Millipore, Bedford, MA, USA). eNOS, phospho-Ser1177-eNOS, iNOS, NF-κB p105, p50, p65, p-IκB, IκB, and β-actin on the membrane were probed with respective primary antibodies and horseradish peroxidase (HRP)-conjugated secondary antibodies (1:5000 dilution; Vector Laboratories, Burlingame, CA, USA). Immunoreactivity was visualized by ECL plus luminal solution (Amersham Biosciences, Piscataway, NJ, USA). The intensities of designated protein bands were densitometrically quantified and normalized with β-actin.

### 4.10. Immunofluorescence Assay

The protocols for immunofluorescence staining of eNOS, phospho-Ser1177 eNOS, iNOS, and NF-κB in HUVECs were performed, as described previously [21]. HUVECs were treated alone or in combination with PBS, 10 nM α-MSH, 10 μM H89, 5 mM l-arginine, 10 μg/mL MC1-R antibody, and 10 μg/mL MC2-R antibody for 24 h. Cell cultures were fixed, permeabilized with 0.1% Triton X-100, blocked in 0.1% normal goat serum and followed by incubating with respective antibodies (1:1000 dilution) at 4 °C for 16 h. After rinsing, the specimens were further incubated with Alexa-488- or Alexa-546-conjugated secondary antibody (1:1000 dilution; Molecular Probes, Thermo Fisher Scientific, Waltham, MA, USA) for 1 h, counterstained with DAPI, and then mounted in anti-Fade media (Invitrogen, Carlsbad, CA, USA). Immunofluorescence images were captured with ZEISS LSM PASCAL multiphoton confocal microscope image system (Carl Zeiss, Oberkochen, Germany).

### 4.11. MC1-R and MC2-R Gene Silencing Assay

MC1-R and MC2-R gene silencing were respectively utilized though transfection with MC1-R and MC2-R siRNA according to the manufacturer’s instructions (Ambion by Life Technologies, Paisley, Scotland, UK). The sequences of the MC1-R siRNA were as follows: sense 5′-CCAGAAAGCUUCAUCCACAtt-3′ and antisense 5′-UGUGGAUGAAGCUUUCUGGtc-3′ (Ambion; Life technologies Ltd., Paisley, Scotland, UK). The sequences of MC2-R siRNA was as the following: sense 5′-CGCUGAUGCUGGUCUUCAUtt-3′ and antisense 5′-AUGAAGACCAGCAUCAGCGgg-3′ (Ambion; Life technologies Ltd., Paisley, Scotland, UK). For each transfection, 0.5 μg of siRNA or scramble siRNA within 4 μL of siRNA transfection reagent was added to siRNA transfection media. The siRNA transfection media was overlaid onto the cells for 6 h. The media was then aspirated and the cells were maintained in M199 medium for the further study. After transfection for 24 h, HUVECs were treated with α-MSH (10 nM) for 24 h subjected to further assays.

### 4.12. NF-κB Activity Assay

The NF-κB activities in HUVECs were measured using luciferase activity protocols [42]. In brief, HUVECs incubated in a six-well plate to reach 80% confluency were co-transfected with *Renilla reniformis* luciferase reporter vector (Promega, Madison, WI, USA) and NF-κB-driven luciferase vector (Stratagene, La Jolla, CA, USA) at a ratio of 2:5 using Lipofectamine (Invitrogen, Carlsbad, CA, USA). After treatment with 10 nM α-MSH either/or combination with 10 ng/mL LPS, 10 μg/mL MC1-R antibody, 10 μg/mL MC2-R antibody, MC1-R and MC2-R siRNA for 24 h, the luciferase activities were detected with Dual-Light kits (Promega, Madison, WI, USA) and the luminometer (Microlumat Plus LB96V; Berthold Technologies, Bad Wildbad, Germany) and normalized with that of *R. reniformis* luciferase, according to the manufacturer’s instructions.

### 4.13. Animal Use and Care Ethics

Protocols for animal use were performed in compliance with the US National Institutes of Health Guidelines for the Care and Use of Laboratory Animals, and approved by the Institutional Animal Care and Use Ethics Committee of National Sun Yat-sen University (approval reference #10305, 2014/05/01-2017/04/30).

### 4.14. Statistical Analysis

All data were expressed as mean ± standard deviation (SD). A one-way ANOVA was applied to assess the statistical differences followed by Newman–Keuls multiple comparison tests among the groups. The differences were considered statistically significant when *p* was less than 0.05.

## 5. Conclusions

In summary, the present study provides evidence that α-MSH suppresses angiogenesis through MC1-R/MC2-R/PKA signaling to inhibit NF-κB signaling and the downstream eNOS/iNOS expression. It results in depleted NO homeostasis and perturbed angiogenic processes in endothelial cells, and ultimately leads to defective vessel formation. Our findings highlight the role of NO/NOSs signaling in the anti-angiogenic function of α-MSH and offer the therapeutic perspective of α-MSH for harmonizing vascular homeostasis for diseases due to excessive angiogenesis. The NO-modulatory function of α-MSH advocates its therapeutic potential for diseases due to aberrant angiogenesis.

## Figures and Tables

**Figure 1 ijms-19-03823-f001:**
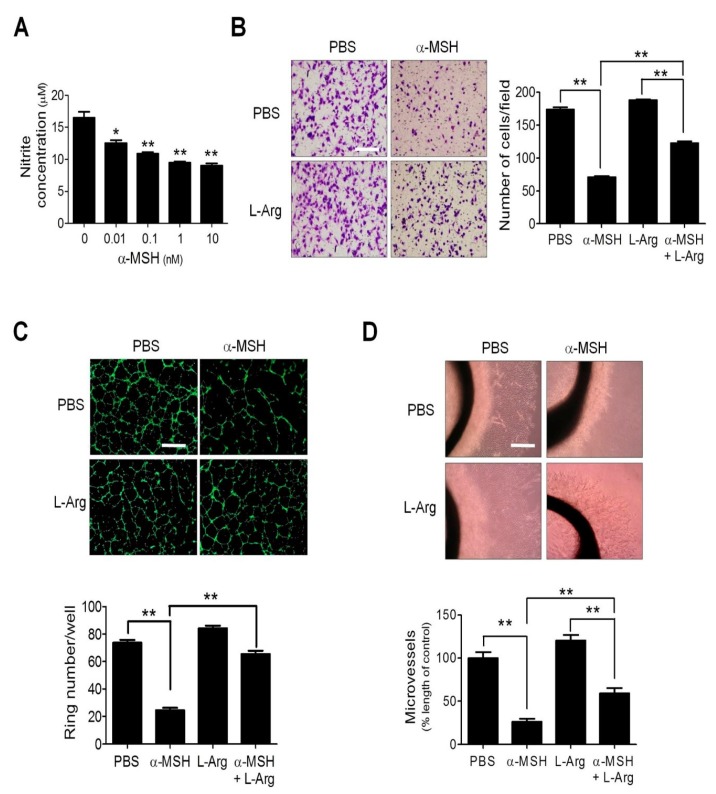
Effects of l-Arginine supply on α-MSH-induced angiogenesis inhibition in endothelial cells and aorta rings. (**A**) Dose-dependent effect α-Melanocyte-Stimulating Hormone (α-MSH) (0.01–10 nM for 24 h) on the nitrite levels in human umbilical vein endothelial cells (HUVECs). (**B**) Effect of l-Arg (5 mM) on migration in α-MSH-treated HUVECs. (**C**) Effect of l-Arg on tube formation in α-MSH-treated HUVECs. (**D**) Effect of l-Arg (5 mM) on microvessel sprouting in α-MSH-treated aortic rings. Data are expressed as mean ± SEM from triplicates. Asterisks indicate statistical significance versus control (** *p* < 0.001; * *p* < 0.005); Scale bars, 100 μm (**B**,**C**) and 2 mm (**D**).

**Figure 2 ijms-19-03823-f002:**
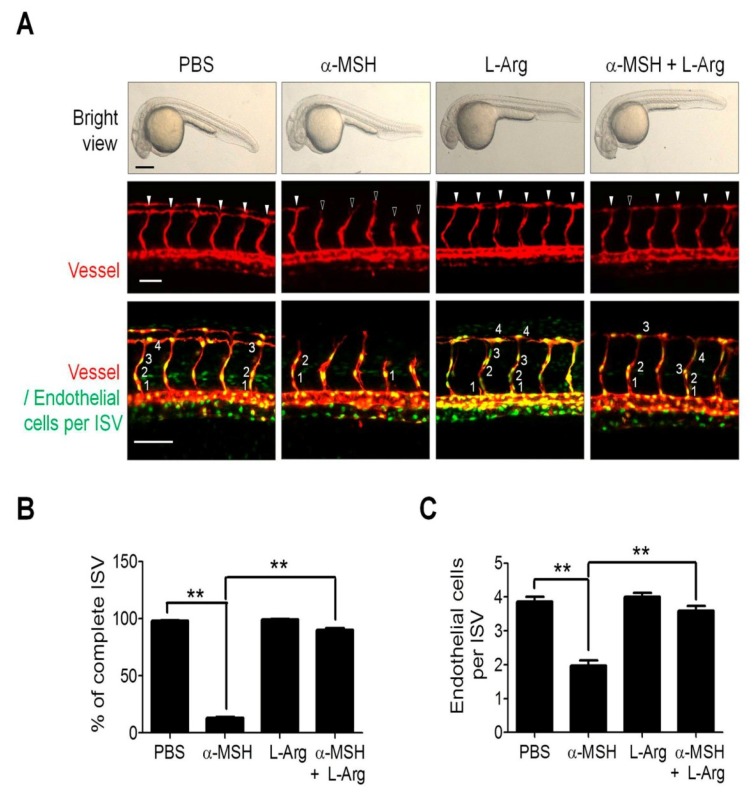
Effect of l-Arginine on vascular development and endothelial recruitment in α-MSH-treated transgenic zebrafish. (**A**) Effect of l-Arg supply (5 mM) on vascular development and endothelial recruitment in α-MSH-treated transgenic zebrafish. Representative confocal images of phenotype and vascular growth of ISV in *Tg*(*kdrl:mCherry^ci5^;fli1a:negfp^y5^*) double transgenic zebrafish embryos. The l-Arg-treated embryos displayed minor responses to the α-MSH inhibition of vessel formation (red fluorescence) and endothelial cell recruitment (green fluorescence in cell nuclei). Solid and hollow arrowheads indicate complete and incomplete vessel formation in ISV. (**B**) Quantification analysis of the ISV completion rate in ISV of α-MSH-treated transgenic zebrafish. (**C**) Quantification analysis of the number of endothelial cells per ISV of α-MSH-treated transgenic zebrafish. Data are expressed as mean ± SEM from 30 ISVs in at least 3 embryos. Asterisks indicated statistical significance versus control (** *p* < 0.001); Scale bars in the upper, middle, and lower panels were 50 μm, 50 μm and 100 μm, respectively.

**Figure 3 ijms-19-03823-f003:**
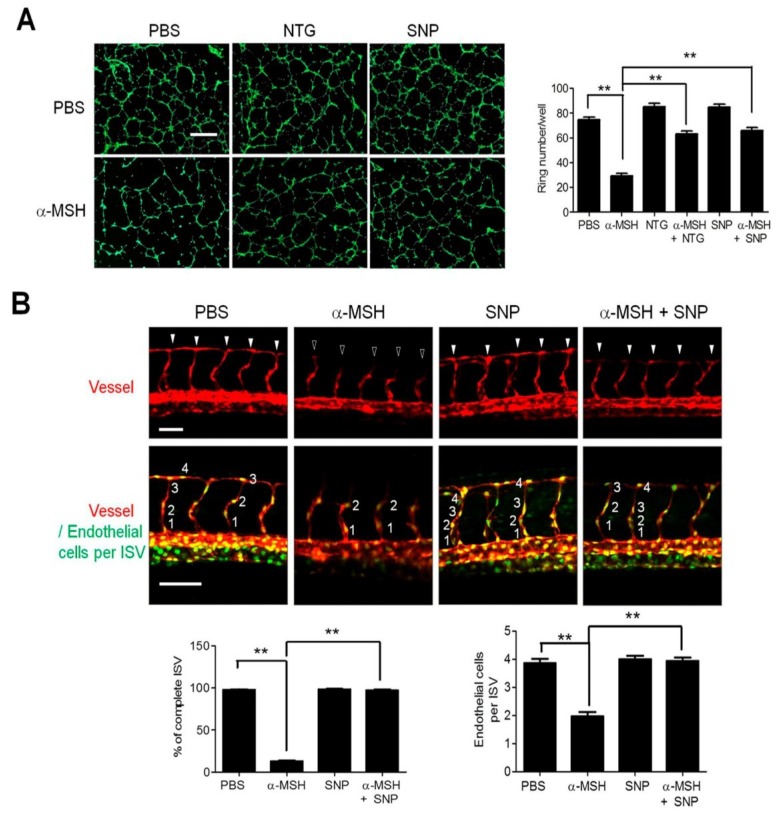
Effect of NO/sGC modulators on the α-MSH-induced angiogenesis in vitro and in vivo. (**A**) Effect of synthetic NO donors (NTG and SNP; 10 μM) on α-MSH-induced inhibition of tube formation in HUVECs. Data are expressed as mean ± SEM from triplicates. Scale bars, 100 μm. (**B**) Effect of SNP supply on vascular development and endothelial recruitment in α-MSH-treated transgenic zebrafish. Confocal images of vessels (red fluorescence) and endothelial cells (green fluorescence) in ISV in the α-MSH-stressed zebrafish embryos after SNP treatment (10 μM). The ISV morphogenesis and endothelial cell number in each ISV were expressed as mean ± SEM calculated from 30 ISVs in at least 3 embryos. Solid and hollow arrowheads indicate complete and incomplete vessel formation in ISV. Scale bars in the upper and lower panels were 50 and 100 μm, respectively. Data were expressed as mean ± SEM from triplicate experiments. Asterisks indicate statistical significance versus control (** *p* < 0.001).

**Figure 4 ijms-19-03823-f004:**
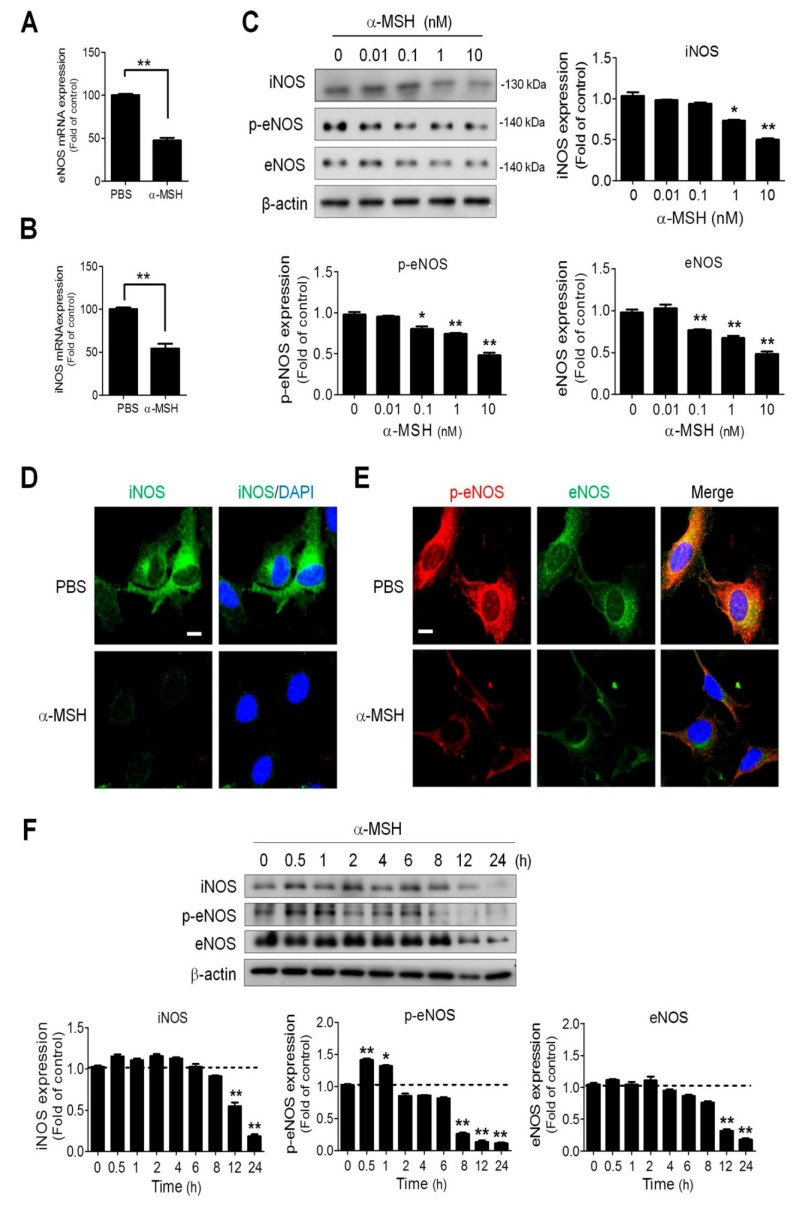
Dose- and time-dependent effect of α-MSH on eNOS phosphorylation and eNOS/iNOS expression in HUVECs. The inhibitory effects of α-MSH on (**A**) eNOS and (**B**) iNOS mRNA expression in HUVECs. (**C**) Dose effect of α-MSH on eNOS phosphorylation and eNOS/iNOS expression in HUVECs. After treatment with α-MSH (0.01–10 nM) for 24 h, the protein extracts of HUVECs were subjected to immunoblot analysis using antibodies against iNOS, phospho-Ser1177-eNOS and eNOS. (**D**) Immunofluorescence analysis of α-MSH treatment (10 nM for 24 h) on iNOS expression (green) in HUVECs. (**E**) Immunofluorescence analysis of α-MSH treatment (10 nM for 24 h) on eNOS phosphorylation (phospho-Ser1177-eNOS in red) and eNOS (in green) expression in HUVECs. The cell nuclei were stained with DAPI (blue). (**F**) Time-dependent effect of α-MSH on eNOS phosphorylation and eNOS/iNOS expression in HUVECs. After treatment with α-MSH (10 nM) for different intervals, the protein extracts of HUVECs were subjected to immunoblot analysis. Data were expressed as mean ± SEM calculated from triplicates. Quantification indicated mean fold change compared with the control after the expression levels were normalized to β-actin levels. Scale bars, 100 μm; * *p* < 0.05 and ** *p* < 0.01 compared with the control groups.

**Figure 5 ijms-19-03823-f005:**
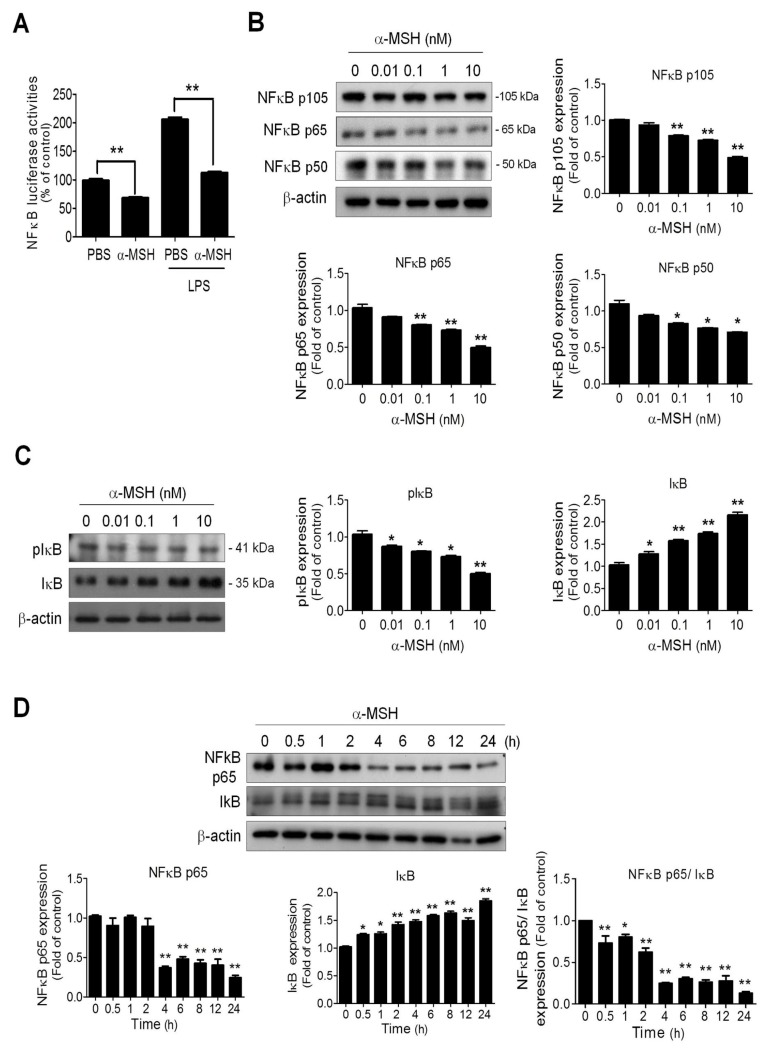
Effect of α-MSH on NF-κB activities and signaling in HUVECs. (**A**) Effect of α-MSH (10 nM for 24 h) on basal and LPS-stimulated NF-κB-driven luciferase activities in HUVECs. (**B**) Dose effect of α-MSH (0.01–10 nM for 24 h) on NF-κB subunits p105, p65, p50 expression in HUVECs. (**C**) Dose effect (0.01–10 nM for 24 h) of α-MSH on phosphorylated IκB and IκB expression in HUVECs. (**D**) Time-dependent effect of α-MSH (10 nM) on NF-κB p65 and IκB expression in HUVECs. After treatment with α-MSH (10 nM) for different intervals, the protein extracts of HUVECs were subjected to immunoblot analysis. Data were expressed as mean ± SEM from triplicates. Quantification indicated mean fold change compared with the control after the expression levels were normalized to β-actin levels. Scale bars, 100 μm. * *p* < 0.05 and ** *p* < 0.01 compared with the control groups.

**Figure 6 ijms-19-03823-f006:**
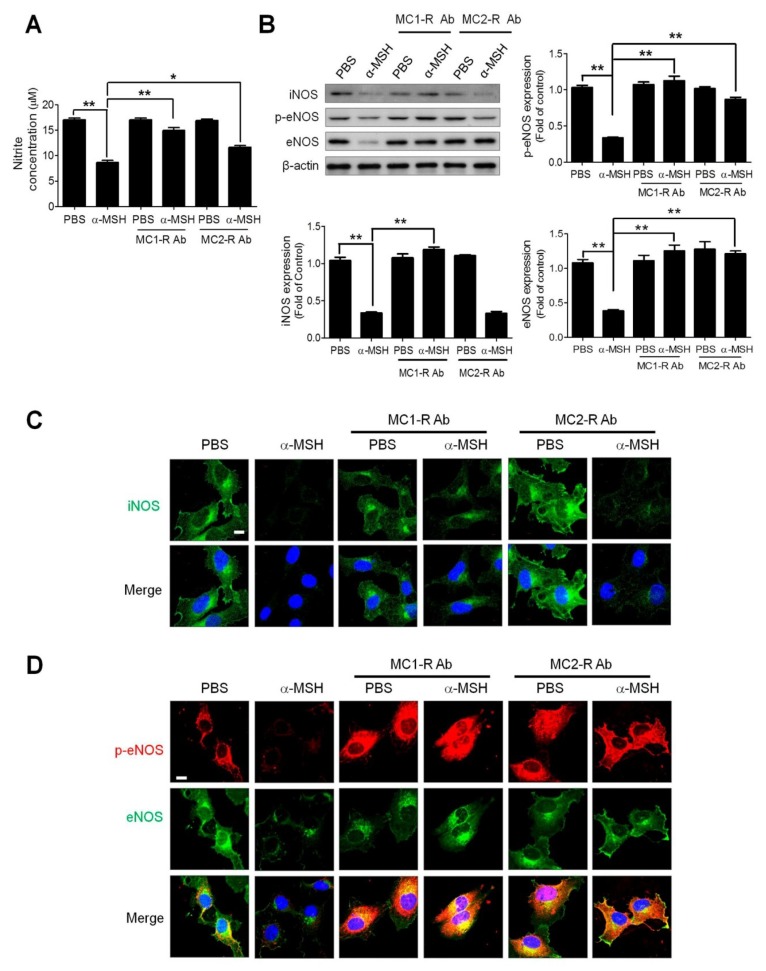
Effect of MC1-R and MC2-R neutralization on NO release and iNOS /eNOS expression in HUVECs. (**A**) Effect of MC1-R and MC2-R neutralization on nitrite production in HUVECs after α-MSH treatment (10 nM for 24 h). (**B**) Immunoblot analysis of the effect of MC1-R and MC2-R neutralization on phospho-Ser1177 eNOS phosphorylation and eNOS/iNOS expression in α-MSH-treated HUVECs. (**C**) Immunofluorescence analysis of the effect of MC1-R and MC2-R neutralization on iNOS expression (green) in α-MSH-treated HUVECs. (**D**) Immunofluorescence analysis of the effect of MC1-R and MC2-R neutralization on phospho-Ser1177-eNOS (red) and eNOS (green) expression. Cell nuclei were counterstained with DAPI (blue). Data are expressed as mean ± SEM calculated from triplicates. Quantification indicated mean fold change compared with the control after the expression levels were normalized to β-actin levels.Scale bars, 100 μm. * *p* < 0.05 and ** *p* < 0.01 compared with the control groups.

**Figure 7 ijms-19-03823-f007:**
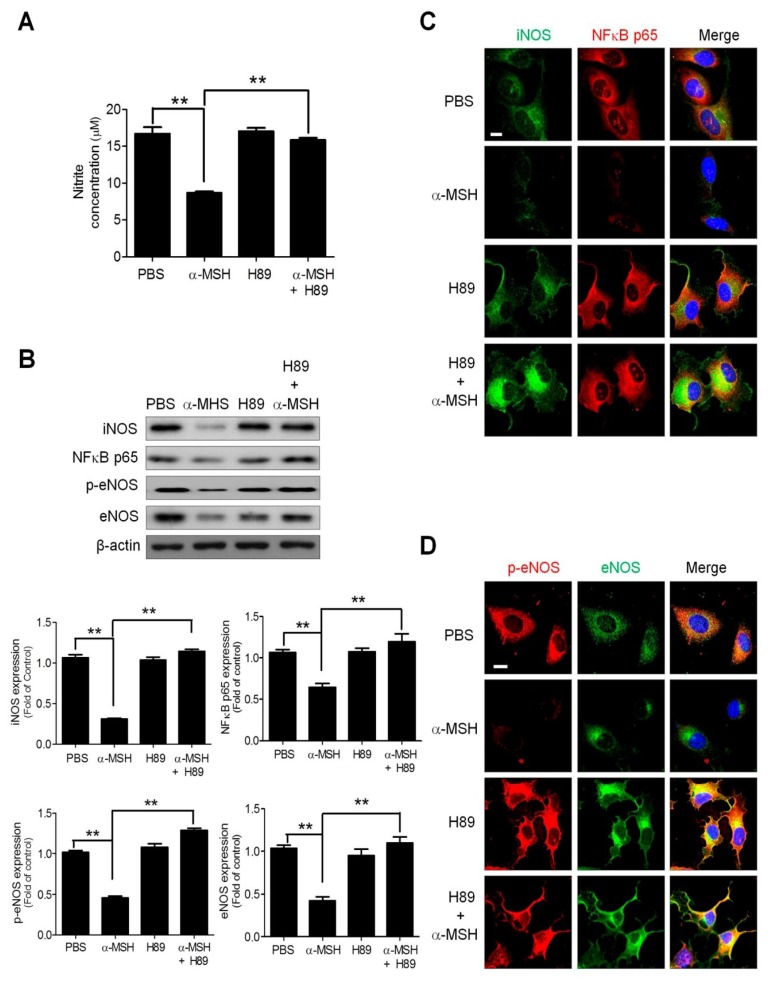
Effects of PKA antagonist H89 on NO homeostasis and eNOS/iNOS expression in α-MSH-treated HUVECs. (**A**) Effect of H89 (10 μM) on nitrite production in HUVECs. (**B**) Immunoblot analysis of the effect of H89 on eNOS, phospho-Ser1177-eNOS, iNOS, and NF-κB p65 expression in α-MSH-treated HUVECs. (**C**) Immunofluorescence analysis of the effect of H89 on iNOS (green) and NF-κB p65 (red) level in α-MSH-treated HUVECs. (**D**) Immunofluorescence analysis of the effect of H89 on phospho-Ser1177-eNOS (red) and eNOS (green) expression in α-MSH-treated HUVECs. Cell nuclei were counterstained with DAPI (blue). Scale bars, 100 μm. Data are expressed as mean ± SEM from triplicates. Quantification indicated mean fold change compared with the control after the expression levels were normalized to β-actin levels. Scale bars, 100 μm. ** *p* < 0.01 compared with the control groups.

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
