# Peer review of "α-Melanocyte-Stimulating Hormone Attenuates Neovascularization by Inducing Nitric Oxide Deficiency via MC-Rs/PKA/NF-κB Signaling"

_ijms, 2018, doi:10.3390/ijms19123823_

Reviewer 1 Report

What is the clinical relevance of anti angiogenic function of MSH? In many circumstances vascularization is an adaptive process with protective effects. Authors should write about it in MS.

Did authors test the MSH+NO inhbitor (L-NAME) effect together? Is there any pubmed data about it?

Author Response

Point 1: What is the clinical relevance of anti-angiogenic function of MSH? In many circumstances vascularization is an adaptive process with protective effects. Authors should write about it in MSH.

Response 1:

Thank for the excellent suggestion. Recent studies have demonstrated the anti-neoplastic potential of pro-opiomelanocortin, the precursor of a-MSH, in various types of cancer through neovascularization blockade (Hum Gene Ther. 22(3):325-35, 2011; J. Gene Med. 14: 44–53, 2012; Mol Cancer Ther. 12(6):1016-25, 2013). Thus, the anti-angiogenic function of a-MSH is highly relevant to clinical science and could be applicable to diseases due to excessive neovascularizations.   

Point 2: Did authors test the MSH+NO inhibitor (L-NAME) effect together? Is there any pubmed data about it? 

Response 2:

Thank for the excellent comment. We did not test the influence of L-NAME on the function of a-MSH in this study. However, regarding the Pubmed search, there were some literatures, including our own (J Pharmacol Exp Ther 321 (2007) 455-461), indicating a-MSH administration induces vessel relaxation ex vivo and hypotension in animals within a short period (less than 10 minutes) through NOSs/NO signaling. The differential effect of a-MSH on NO homeostasis is elaborated in the Discussion section of the revision (paragraph 2; line 317-333).

Reviewer 2 Report

The manuscript α-Melanocyte-Stimulating Hormone Attenuates Neovascularization by Inducing Nitric Oxide Deficiency via MC-Rs/PKA/NFκB Signaling by Weng et al is an interesting report on the presented study.

The paper is well written in general and requires some minor spell checking (for instance line 107 "supplement" should probably be "supplementation".

With regards to the content, I miss the discussion why L-arginine has been used as a NO donor in this study. In recent years studies have been published on the beneficial effects of L-citrulline over L-arginine in fields of sepsis, angiogenesis, inflammation etc.

In paragraph 2.4 the results from figure 4 are discussed. In fig. 4c, expression of p-ENOS and eNOS is presented with similar decreasing expression with higher concentrations of MSH. If the total eNOS is lower than lower p-eNOS is an obvious result. Please calculate the ratios between both.

How does nNOS play a role in your proposed mechanisms? From research into bone healing it is known that it has no acute effects for instance. If this also holds in your study please indicate so.

The figure quality of some blots (for instance in fig 5c) is low, try to improve this.

Author Response

Response to Reviewer 2 Comments

Point 1: The paper is well written in general and requires some minor spell checking (for instance line 107 "supplement" should probably be "supplementation". 

Response 1:  

Thank for the kind suggestion. The word “supplement” is corrected as “supplementation”. We have re-edited the entire manuscript to minimize such errors.

Point 2: With regards to the content, I miss the discussion why L-arginine has been used as a NO donor in this study. In recent years studies have been published on the beneficial effects of L-citrulline over L-arginine in fields of sepsis, angiogenesis, inflammation etc. 

Response 2:

Thanks for the thoughtful comment. L-arginine is converted in the body into nitric oxide and causes blood vessels to open wider for improved blood flow by nitric oxide. Besides, l-arginine availability has been confirmed to modulate local nitric oxide (Infect Immun. 2000;68(8):4653-7).

Point 3: In paragraph 2.4 the results from figure 4 are discussed. In fig. 4c, expression of p-ENOS and eNOS is presented with similar decreasing expression with higher concentrations of MSH. If the total eNOS is lower than lower p-eNOS is an obvious result. Please calculate the ratios between both.

Response 3:  

Thanks for the comment. By calculating the ratios between p-eNOS and eNOS as an index of eNOS activities, it was also found that a-MSH elicited a dose-dependent reduction of p-eNOS/eNOS at high concentration in HUVEC (as shown below).

The dose-dependent effect of α-MSH on the levels of eNOS activities in HUVEC.  The inhibitory effects of a-MSH on eNOS activities in HUVECs. After treatment with a-MSH (0.01-10 nM) for 24 h, the protein extracts of HUVECs were subjected to immunoblot analysis using antibodies against phospho-Ser1177-eNOS and eNOS. Data are expressed as mean ± SEM calculated from triplicate experiments which were repeated at   least three times. Asterisks indicate statistical significance compared with control group (**p < 0.001; *p < 0.005);

Point 4: How does nNOS play a role in your proposed mechanisms? From research into bone healing it is known that it has no acute effects for instance. If this also holds in your study please indicate so.

Response 4:

Thanks for insightful comments. However, we did not observe the role of nNOS in this study. It will be interesting to dissect whether nNOS play a role in a-MSH- mediated anti-angiogenesis in our future studies.

Point 5: The figure quality of some blots (for instance in fig 5c) is low, try to improve this.

Response 5:

Thanks for the kind suggestion. We have improved the blots quality in all figures of the revision.